

**Formation of secondary organic aerosols from gas–phase**
**emissions of heated cooking oils**
Tengyu Liu[1], Zijun Li[2], ManNin Chan[2,3], and Chak K. Chan[1,*]
1. School of Energy and Environment, City University of Hong Kong, Hong Kong,
China.
2. Earth System Science Programme, The Chinese University of Hong Kong, Hong
Kong, China.
3. The Institute of Environment, Energy and Sustainability, The Chinese University of
Hong Kong, Hong Kong, China
*Corresponding author:
Dr. Chak K. Chan
School of Energy and Environment, City University of Hong Kong
Tel: +852-34425593
Email: Chak.K.Chan@cityu.edu.hk



**Abstract**
Cooking emissions can potentially contribute to secondary organic aerosol (SOA) but
remain poorly understood. In this study, formation of SOA from gas-phase emissions
of five heated vegetable oils (i.e. corn, canola, sunflower, peanut and olive oils) was
investigated in a potential aerosol mass (PAM) chamber. Experiments were conducted
at 19-20 ℃ and 65-70% RH. The characterization instruments included a scanning
mobility particle sizer (SMPS) and a high-resolution time-of-flight aerosol mass
spectrometer (HR-TOF-AMS). The efficiency of SOA production, in ascending order,
was peanut oil, olive oil, canola oil, corn oil and sunflower oil. The major SOA
precursors from heated cooking oils were related to the content of mono-unsaturated
fat and omega-6 fatty acids in cooking oils. The average production rate of SOA, after
aging at an OH exposure of $1.7 \times 10^{11}$ molecules cm$^{-3}$ s, was $1.35 \pm 0.30$ µg min$^{-1}$, three
orders of magnitude lower compared with emission rates of fine particulate matter
(PM$_{2.5}$) from heated cooking oils in previous studies. The mass spectra of cooking SOA
highly resemble field-derived COA (cooking-related organic aerosol) in ambient air,
with $R^2$ ranging from 0.74 to 0.88, suggesting that COA might not be entirely primary
in origin. The average carbon oxidation state (OS$_c$) of SOA was -1.51 – -0.81, falling
in the range between ambient hydrocarbon-like organic aerosol (HOA) and semi-
volatile oxygenated organic aerosol (SV-OOA), indicating that SOA in these
experiments was lightly oxidized.





## 1. Introduction

Organic aerosol (OA) is an important component of atmospheric particulate matter

(PM), which influences air quality, climate and human health (Hallquist et al., 2009). A

significant fraction of OA is secondary organic aerosol (SOA) (Zhang et al., 2007),

formed via the oxidation of volatile organic compounds (VOCs) (Hallquist et al., 2009).

However, chemical transport models generally underestimate SOA levels due to the

unclear sources and formation processes of SOA (de Gouw et al., 2005; Heald et al.,

2005; Johnson et al., 2006; Volkamer et al., 2006). Recently, primary semi-volatile and

intermediate-volatility organic compounds (SVOCs and IVOCs) that can come from

the evaporation of primary organic aerosol (POA) were found to form substantial SOA

(Robinson et al., 2007; Donahue et al., 2009). Therefore, any source of POA may be

associated with the production of SOA.

Cooking-related organic aerosol (COA), thought to be primary in origin,

contributed 10–34.6% of the total OA in urban areas (Allan et al., 2010; Sun et al., 2011;

2012; Ge et al., 2012; Mohr et al., 2012; Crippa et al., 2013; Lee et al., 2015). Lee et al.

(2015) found that COA even dominated the contribution to POA at roadside sites in the

commercial and shopping area of Mongkok in Hong Kong. Cooking may be a large

source of SOA in urban areas, yet the formation of SOA from cooking remains poorly

understood. Kaltsonoudis et al. (2016) observed that the oxygen to carbon ratio (O:C)

of OA from meat charbroiling increased from 0.09 to 0.30 after a few hours of chemical

aging. The aged aerosol mass spectra have similarities with ambient COA factors in

two major Greek cities. Hayes et al. (2015) modeled that cooking emissions contributed



19–35% of SOA mass in downtown Los Angeles during the California Research at the
Nexus of Air Quality and Climate Change (CalNex) 2010 campaign. In their study,
primary SVOCs and IVOCs from cooking emissions were modeled using the same
parameters as those from vehicle exhaust, due to limited information about SOA
formation from cooking (Hayes et al., 2015).
Heating cooking oils, a fundamental process of frying, was found to produce large
amounts of fine particulate matter ($PM_{2.5}$) (Amouei Torkmahalleh et al, 2012; Gao et
al., 2013) and VOCs (Katragadda et al., 2010; Klein et al., 2016a). The $PM_{2.5}$ emission
rate for peanut, canola, corn and olive oils heated at 197 °C was shown to be as high as
54 mg $min^{-1}$ (Amouei Torkmahalleh et al, 2012). Allan et al. (2010) found that cooking
oils may contribute more to PM than the meat itself in urban areas of London and
Manchester. The VOCs emitted from heated cooking oils were dominated by aldehydes
(Klein et al., 2016a), which were suggested to be potential SOA precursors (Chacon-
Madrid et al., 2010). Despite these previous efforts, there are still no available data
regarding SOA formation from heated cooking oils.
The objective of this study is to characterize SOA formation from gas–phase
emissions of heated cooking oils. The magnitude and composition of the SOA formed
from gas–phase emissions of heated cooking oils were evaluated and have been
discussed for the first time in this paper.
**2. Materials and methods**
**2.1 PAM chamber**



SOA formation from gas–phase emissions of five different heated cooking oils was
investigated in a potential aerosol mass (PAM) chamber, which has been described in
detail elsewhere (Kang et al., 2007, 2011; Lambe et al., 2011, 2015). Briefly, a PAM
chamber is a continuous flow stainless steel cylindrical reactor using high and
controlled levels of oxidants to oxidize precursor gases to produce SOA. The volume
is approximately 19 L (length 60 cm, diameter 20 cm). High OH exposures were
produced through the photolysis of ozone irradiated by a UV lamp ($\lambda = 254$ nm) in the
presence of water vapor. Ozone was produced by an ozone generator (1000BT-12,
ENALY, Japan) via irradiation of pure $O_2$. The OH concentration was controlled by the
flow rate of ozone in the PAM chamber, which was approximately 40 ppm prior to
dilution. The total flow rate in the PAM chamber was set at 3 L min$^{-1}$ by a mass flow
controller, resulting in residence time of 380 s. The corresponding upper limit of OH
exposure at these operating conditions was $1.7 \times 10^{11}$ molecules cm$^{-3}$ s, which is
equivalent to 1.3 days of atmospheric oxidation, assuming an ambient OH
concentration of $1.5 \times 10^6$ molecules cm$^{-3}$ (Mao et al., 2009). The upper limit of OH
exposure was determined by measuring the decay of $SO_2$ (Model T100, TAPI Inc, USA),
following previous procedures (Kang et al., 2007; Lambe et al., 2011). Before and after
each experiment, the PAM reactor was cleaned by exposure to a high concentration of
OH until the mass concentration of background particles was less than 5 µg m$^{-3}$.

The PAM chamber was designed with a large radius and a small surface-to-volume

ratio to minimize wall effects. The transmission efficiency for particles at a mean
mobility diameter ($D_m$) larger than 150 nm was greater than 80% (Lambe et al., 2011).





The wall loss of particles was considered to be small, as the particles larger than 150
nm accounted for greater than 70% of the aerosol mass (Fig. S1 in the supporting
information). Transmission efficiency of gases in the PAM chamber indicates that vapor
wall losses in the PAM chamber are negligible (Lambe et al., 2011).
**2.2 Experimental conditions**
A schematic of the experimental setup is shown in Fig. 1. The tested vegetable oils,
purchased from a local supermarket, included canola, corn, sunflower, peanut and olive
oils. For each experiment, 30 mL vegetable oil was heated at approximately 220 ℃ for
20 min in a 500 mL Pyrex bottle on an electric heating plate. Prior to introduction to
the PAM chamber, particles from the heated oil emissions were removed using a Teflon
filter. A 2 m Teflon tube was used as the transfer line to minimize wall loss of VOCs.
After 10 min of heating, the UV lamp was turned on and the emissions were exposed
to high OH levels for approximately one hour. Once the UV lamp was turned off, the
PAM reactor was flushed continuously using pure $N_2$ and $O_2$ until the aerosol mass was
below 3 μg m$^{-3}$. Then the experiment was repeated at another OH level. The RH and
temperature of the PAM outflow were measured continuously (HMP 110, Vaisala Inc,
Finland) and stabilized at 65-70% and 19-20 ℃, respectively. The adjustment of RH
was achieved by passing the pure $N_2$ and $O_2$ through water bubblers. Blank experiments
were conducted in the absence of cooking oils under similar conditions to quantify the
amount of aerosols formed from matrix gas when exposed to different OH levels.
POA emitted from heated cooking oils was also characterized in this study. For
each test, 30 mL vegetable oil was heated to 240 ℃ for 2 min in a pan on an induction



cooker. The emissions, after passing through a mixing chamber of 36 L, were
introduced to the PAM chamber by a Dekati diluter (Dekati Ltd, Finland) at a flow rate
of 0.15 L min$^{-1}$, achieving a final dilution ratio of approximately 160. No ozone was
introduced to the PAM chamber during measurement, and the UV lamp was off.
Temperature and RH were similar to those of the SOA formation experiments.
A scanning mobility particle sizer (SMPS, TSI Incorporated, USA, classifier
model 3082, CPC model 3775) was used to measure particle number concentrations
and size distributions. Particle size ranged from 15 nm to 661 nm. An aerosol density
of 1.4 g cm$^{-3}$ was assumed to estimate the SOA mass from the particle volume
concentration (Zhang et al., 2005). For the SOA formation experiments, the
contribution from background organic aerosols was subtracted from the total organic
aerosols. The maximum concentration of background organic aerosols was 8.4 μg m$^{-3}$,
almost negligible compared with the dozens to several hundreds of μg m$^{-3}$ of SOA
formed in this study. The organic aerosol composition was characterized by a high-
resolution time-of-flight aerosol mass spectrometer (HR-TOF-AMS, abbreviated as
AMS hereafter, Aerodyne Research Incorporated, USA) (DeCarlo et al., 2006). A
diffusion dryer was connected to the sampling line to remove water. The instrument
was operated in the high sensitivity V-mode and high resolution W-mode alternating
every one minute. The toolkit Squirrel 1.57I and Pika 1.16I were used to analyze the
AMS data. The molar ratios of hydrogen to carbon (H:C) and oxygen to carbon (O:C)
were determined with the improved-ambient method (Canagaratna et al., 2015). The
ionization efficiency of AMS was calibrated using 300 nm ammonium nitrate particles.



The particle-free matrix air, obtained by passing the air through a HEPA filter, was
measured for at least 20 min before each experiment to determine the signals from
major gases. The collection efficiency (CE) was corrected by comparing AMS mass
concentrations to concurrent SMPS mass concentrations, following the methods of
Gordon et al. (2014) and Liu et al., (2015).
**2.3 SOA production rate**
The SOA production rate (PR) was expressed as micrograms (µg) of SOA produced
per minute (min), calculated using the following equation, similar to calculation of
emissions rates of primary particles from cooking (Klein et al., 2016a):
$$PR = [SOA] \times DR \times F \qquad (1)$$
where [SOA] is the SOA concentration in $\mu g\ m^{-3}$; DR is the dilution ratio and F is the
flow rate in $m^3\ min^{-1}$ of the carrier gas. All gas-phase emissions from heated cooking
oils were assumed to be transported into the PAM chamber.

Emission rates are commonly used to normalize PM emissions from cooking

activities (Amouei Torkmahalleh et al, 2012; Gao et al., 2013; Klein et al., 2016a, b).
Here, the adoption of SOA PR, similar to emission rates, facilitates the normalization
of SOA production from cooking and direct comparison of the amount of primary
emitted and secondary formed particles. Though SOA yields were not determined due
to the lack of VOC concentrations, we believe that SOA PR is a useful metric for the
estimation of SOA production from cooking and can be used for comparison among
different studies.
**3.  Results and discussion**





### 3.1 SOA formation


In Fig. 2, we plot the time series of RH, ozone and organic aerosol concentrations during
the aging of gas–phase emissions from heated peanut oil. As described above, the ozone
concentration prior to dilution was stable at approximately 40 ppm. The pulse of RH
was caused by disconnection of the introduction line when changing the Teflon filter.
During the initial 10 min of heating, the mass concentration of organics was close to
the detection limit of the instrument, indicating that POA emissions were thoroughly
removed by the Teflon filter. Immediately after oxidation was initiated by turning on
the UV lamp, substantial SOA was formed, and its concentration stabilized after about
20 min. The SOA concentration subsequently reported is the average for the steady
period.
Fig. 3 shows SOA concentration as a function of OH exposure and photochemical
age in days during the aging of gas–phase emissions from different heated cooking oils.
The OH exposure ranged from $2.7 \times 10^{10}$ molecules cm$^{-3}$ s to $1.7 \times 10^{11}$ molecules cm$^{-3}$ s,
corresponding to 0.2–1.3 days of photochemical age, assuming 24 h average ambient
OH concentrations of $1.5 \times 10^6$ molecules cm$^{-3}$ (Mao et al., 2009). For all experiments,
the SOA concentration almost linearly increased from 41–107 μg m$^{-3}$ to 320–565 μg m$^{-}$
$^3$ as OH exposure increased. This linear increase has also been observed from vehicle
exhaust at a similar range of OH exposures (Tkacik et al., 2014). Typically, VOCs are
oxidized through functionalization reactions to produce less volatile organics that
readily condense to form SOA. Upon further oxidation, fragmentation reactions and
cleavage of carbon bonds can occur and form more volatile products that reduce SOA



levels (Kroll et al., 2009). In this study, functionalization reactions dominated SOA
formation as reflected by the increase of SOA concentrations shown in Fig. 3.

The slope of the fitted straight line to the SOA data was calculated to estimate the

efficiency of different cooking oils in producing SOA (Table 1). The efficiency of SOA
production, in ascending order, was peanut oil, olive oil, canola oil, corn oil and
sunflower oil. The slope of sunflower oil was $3.82 \times 10^{-15} \, \mu g \, molecules^{-1} \, s^{-1}$, more than
two times that of peanut oil. The different slopes might be related to the emission rate
and composition of VOCs from various cooking oils. Table 1 presents the type of fat
content of the different cooking oils. Unsaturated fat accounts for 75%-88% of the total
fat content. A multivariate linear regression was used to relate the SOA production
efficiency to the fat content of cooking oils. The intercept was set to zero. The resulting
equation was $Y = 2.62 \times 10^{-17} X_1 + 4.71 \times 10^{-17} X_2$, where Y is the SOA production
efficiency ($\mu g \, molecules^{-1} \, s^{-1}$); $X_1$ and $X_2$ represent the content of mono-unsaturated fat
(%) and omega-6 fatty acid (%) in cooking oil, respectively. The SOA production
efficiency was strongly correlated ($R^2 = 0.97$, $p<0.05$) with the content of mono-
unsaturated fat and omega-6 fatty acids. This indicated that the major SOA precursors
from heated cooking oils were related to the content of mono-unsaturated fat and
omega-6 fatty acids in cooking oils. Moreover, omega-6 fatty acids dominated the
contribution to SOA production. Omega-6 fatty acids are a family of poly-unsaturated
fatty acids that have in common a final carbon-carbon double bond in the n-6 position,
counting from the methyl end (Simopoulos, 2002). The peroxyl radical reactions of



omega-6 fatty acids might emit long-chain aldehydes (Gardner, 1989), which have been
suggested as potential SOA precursors (Chacon-Madrid et al., 2010).

The average SOA PR from gas–phase emissions of the five cooking oils at an OH

exposure of $1.7 \times 10^{11}$ molecules $cm^{-3}$ s was calculated to be $1.35 \pm 0.30 \, \mu g \, min^{-1}$. Amouei
Torkmahalleh et al. (2012) found that primary $PM_{2.5}$ emission rates for peanut, canola,
corn and olive oils heated at 197 °C ranged from 3.7 mg $min^{-1}$ to 54 mg $min^{-1}$. He et al.
(2004) reported a $PM_{2.5}$ emission rate for frying in vegetable oils of $2.68 \pm 2.18$ mg $min^{-}$
$^{1}$. The SOA PR determined in this study was negligible compared with primary $PM_{2.5}$
emission rates for heated cooking oils and frying in vegetable oils. However, our results
may underestimate SOA production from cooking under real-world conditions. First,
recent studies have demonstrated that the oxidation of IVOCs and SVOCs evaporated
from POA could produce significant SOA (Donahue et al., 2006; Jimenez et al., 2009).
In this study, POA from heated cooking oils was filtered. Second, emissions of SOA
precursors will be enhanced when cooking food compared with heating cooking oils
alone. For instance, long-chain aldehyde emissions from frying processes can be 10
times those of heated oil (Klein et al., 2016a). Large amounts of monoterpenes will be
emitted when frying vegetables or cooking with herbs and black pepper (Klein et al.,
2016a, b). These enhanced emitted precursors may significantly enhance SOA
production. Finally, laboratory and tunnel studies indicate that SOA production from
typical precursors and vehicle exhaust peak at OH exposures higher than $5.0 \times 10^{11}$
molecules $cm^{-3}$ s (Tkacik et al., 2014; Lambe et al., 2015). The relatively lower OH



exposures in this study compared with typical conditions in the atmosphere may lead
to the underestimation of cooking SOA.
**3.2 Mass spectra of POA and SOA**
Fig. 4 shows high-resolution mass spectra of POA and SOA at an OH exposure of
$2.7 \times 10^{10}$ molecules $cm^{-3}$ s from heated canola oil. Other oils have similar mass spectra,
as reflected in the good correlations shown in Table 2. The mass concentration of POA
was approximately 35 µg $m^{-3}$ for canola oil. The prominent peaks in POA from canola
oil were m/z 41 and 55, followed by m/z 29 and 43. The m/z 41, 43 and 55 were
dominated by $C_3H_5^+$, $C_3H_7^+$ and $C_4H_7^+$ ion series, consistent with the previous
observation by Allan et al. (2010). The m/z 29 was instead dominated by ion $CHO^+$,
which can be used as a tracer for organic compounds with alcohol and carbonyl
functional groups, as a result of thermal decomposition of the oils (Lee et al., 2012).
For the SOA mass spectra, the dominating peaks were m/z 28 and 29, followed by m/z
43 and 44. The m/z 28, 29, 43 and 44 were dominated by $CO^+$, $CHO^+$, $C_2H_3O^+$ and
$CO_2^+$, respectively. For all cooking oils, the mass fractions of m/z 28 and 44 in SOA
were higher, while the mass fractions of m/z 55 and 57 in SOA were lower than those
of the corresponding POA. The increase of mass fractions of the oxygen-containing
ions in SOA mass spectra indicated the formation of oxidized organic aerosols.

The correlation coefficients ($R^2$) between POA and SOA unit mass resolution

(UMR) spectra of heated oil and COA resolved by positive matrix factorization (PMF)
analysis (Lee et al., 2015) were calculated and summarized in Table 2 to evaluate their
similarities. The POA mass spectra between different cooking oils exhibited strong





correlations ($R^2$>0.97) and agreed well with the ambient COA factor obtained at
roadside sites in the commercial and shopping area of Mongkok in Hong Kong (Lee et
al., 2015). The SOA mass spectra between different cooking oils displayed good
correlations ($R^2$>0.94), suggesting a high degree of similarity. The mass spectra of
cooking SOA also greatly resemble POA and field-derived COA in ambient air, with
$R^2$ ranging from 0.74 to 0.88. This similarity between the cooking SOA and ambient
COA suggests that the COA resolved based on ambient data may be a convolution of
POA and SOA, even though vegetable oil may not be the oil commonly used in
commercial kitchens. Kaltsonoudis et al. (2016) also observed that the ambient COA
factor in two major Greek cities in spring and summer strongly resembled the aged
SOA from meat charbroiling in a smog chamber.

Fragments derived from the AMS data have been extensively used to explore the

bulk compositions and properties of ambient organic aerosols (Zhang et al., 2005; Ng
et al., 2010; Heald et al., 2010). Here, we use the approach of Ng et al. (2010) by
plotting the fractions of the total organic signal at m/z 43 ($f_{43}$) vs. m/z 44 ($f_{44}$). The m/z
43 signal is abundant in $C_3H_7^+$ and $C_2H_3O^+$ ions, indicating fresh, less oxidized organic
aerosols. The m/z 44 signal, usually dominated by $CO_2^+$ and formed from the thermal
decarboxylation of organic acids, is an indicator of highly oxygenated organic aerosols
(Ng et al., 2010).

In Fig. 5, we plot $f_{43}$ vs. $f_{44}$ of cooking SOA and SOA data from gasoline (Presto

et al., 2014; Liu et al., 2015) and diesel (Presto et al., 2014) vehicle exhaust measured
in a smog chamber, together with the triangle defined by Ng et al. (2010) based on the



analysis of ambient AMS data. The ambient low-volatility oxygenated OA (LV-OOA)
and semi-volatile OOA (SV-OOA) factors fall in the upper and lower regions of the
triangle, respectively. Ng et al. (2010) proposed that aging would converge the $f_{43}$ and
$f_{44}$ toward the triangle apex ($f_{43} = 0.02$, $f_{44} = 0.30$). In this study, the $f_{43}$ and $f_{44}$ ranged
from 0.06 to 0.10 and from 0.05 to 0.07, respectively; they mainly lie in the lower
portion of the SV-OOA region. As shown in Fig. 5, SOA from gasoline and diesel
vehicle exhaust at a similar range of OH exposures had $f_{44}$ values of 0.11–0.12.
Compared with vehicle exhaust, SOA formed from gas-phase emissions of heated
cooking oils was less oxidized. The potential SOA precursors from heated cooking oils
might be long-chain aldehydes, which are less volatile than SOA precursors such as
aromatics and long-chain alkanes from vehicle exhaust. A single polar moiety of first-
generation products from long-chain aldehydes will have low enough volatility to
condense, while more volatile aromatics and long-chain alkanes require more
functionalization to form SOA (Donahue et al., 2012). Therefore, SOA formed from
heated cooking oils was less oxidized. For each cooking oil, there was little change in
$f_{44}$ and a slight increase in $f_{43}$ as OH exposure increased. The increased SOA mass may
facilitate the partitioning of more volatile organics, leading to a slight increase in $f_{43}$
and little change in $f_{44}$. This is consistent with the observation of previous studies that
the $f_{44}$ of SOA from aromatics and monoterpenes varied little and that $f_{43}$ increased
slightly for SOA mass loadings higher than 100 μg m$^{-3}$ (Ng et al., 2010; Kang et al.,

2011).

**3.3 Chemical composition of SOA**



The O:C ratio and the estimated average carbon oxidation state ($OS_c$) ($OSc \approx 2 \times O:C -$
$H:C$) (Kroll et al., 2011) can be used to evaluate the degree of oxidation of organic
aerosols. Fig. 6 shows the evolution of O:C ratios and $OS_c$ of SOA from heated cooking
oils as a function of OH exposure, together with the POA data. The O:C ratios and $OS_c$
of POA were in the range of 0.14–0.23 and -1.61 – -1.44, respectively, comparable to
those of POA from meat charbroiling (Kaltsonoudis et al., 2016). As shown in Fig. 6,
for each cooking oil, the O:C and $OS_c$ of SOA displayed similar trends, initially
decreasing rapidly and then increasing slowly or leveling off (for canola oil only). In
this study, the increased SOA mass loadings led to the rapid decrease of the oxidation
degree when the OH exposure increased from $2.7 \times 10^{10}$ molecules cm$^{-3}$ s to $6.4 \times 10^{10}$
molecules cm$^{-3}$ s. As OH exposure and the resulting OA mass loadings further increase,
even less oxidized and more volatile organics partition into the particle phase and thus
decrease the oxidation degree (Donahue et al., 2006). The difference in O:C for
different cooking oils at the same OH exposure may be attributed to the differences in
gas-phase SOA precursors. In general, the O:C ratios of SOA formed from gas–phase
emissions of heated cooking oils ranged from 0.24 to 0.46 at OH exposures of $2.7 \times 10^{10}$
$- 1.7 \times 10^{11}$ molecules cm$^{-3}$ s. The $OS_c$ of cooking SOA was -1.51 – -0.81, falling in the
range between ambient hydrocarbon-like organic aerosol (HOA, $OS_c = -1.69$) and SV-
OOA ($OS_c = -0.57$) corrected by the improved-ambient method (Canagaratna et al.,
2015). As suggested by Canagaratna et al. (2015), the $OS_c$ is more robust than the $f_{43}/f_{44}$
relationship for evaluating the oxidation degree of organic aerosols, as the former has
been estimated based on the full spectra.



In Fig. S2 we plot the H:C and O:C molar ratios of POA and SOA from heated
cooking oils on a Van Krevelen diagram. The cooking data fell along a line with a slope
of approximately 0, suggesting the chemistry of SOA formation in this study was
alcohol/peroxide formation (Heald et al., 2010; Ng et al., 2011). This slope is different
from ambient OA data of -0.8 determined by the improved-ambient method (Heald et
al., 2010). It is also different from vehicle exhaust data with slopes ranging from -0.59
to -0.36 (Presto et al., 2014; Liu et al., 2015).
**4.  Conclusions**
Formation of SOA from gas-phase emissions of heated cooking oils was investigated
in a PAM chamber at OH exposures of $2.7 \times 10^{10}$ molecules cm$^{-3}$ s to $1.7 \times 10^{11}$ molecules
cm$^{-3}$ s. The $OS_c$ and $f_{43}/f_{44}$ relationship indicated that the SOA formed was lightly
oxidized. The mass spectra of SOA highly resembled POA from heated cooking oils
and COA factors in ambient air. These similarities indicated that ambient COA factors
identified by AMS could contain cooking SOA. The major SOA precursors from heated
cooking oils were related to the content of mono-unsaturated fat and omega-6 fatty
acids in cooking oils. Considering that animal fats such as pork and chicken fat are also
abundant in mono-unsaturated fat and omega-6 fatty acids, gas-phase emissions from
cooking animal fat can be as efficient as vegetable oils in producing SOA. It is
important to note that the reported SOA data only related to gas-phase emissions from
heated cooking oils. The large amounts of POA emitted from cooking oils may also
form SOA after photochemical aging. More work is needed to investigate SOA
formation from emissions of cooking oils and food. In addition, gas-phase SOA



precursors were not characterized and therefore provided limited information on SOA
yields from cooking; we recommend that future work validate our results and perform
similar experiments, with gas-phase emissions measured.

**Acknowledgments**
The work described in this paper was partially sponsored by Project No. 41675117,
supported by the National Natural Science Foundation of China, and was partially
supported by the Shenzhen Research Institute, City University of Hong Kong. Li, Z.
and Chan, M. N. are supported by a Direct Grant for Research (4053159), The Chinese
University of Hong Kong.



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





**Table 1.** SOA production efficiency and type of fat content (%) [a] of different cooking
oils.

| | Slope [b] | Saturated | Mono- | Poly-unsaturated (%) | | Others |
|---|---|---|---|---|---|---|
| | µg molecules$^{-1}$ s$^{-1}$ | (%) | unsaturated (%) | Omega-6 | Omega-3 | (%) |
| sunflower | $3.82\times10^{-15}$ | 10 | 19 | 64 | 0 | 7 |
| corn | $3.31\times10^{-15}$ | 12 | 24 | 56 | 1 | 7 |
| canola | $2.68\times10^{-15}$ | 7 | 59 | 20 | 9 | 5 |
| olive | $2.55\times10^{-15}$ | 13 | 71 | 8 | 1 | 7 |
| peanut | $1.7\times10^{-15}$ | 16 | 44 | 31 | 0 | 9 |

[a] The type of fat content of cooking oils was derived from skillsyouneed.com.
[b] SOA production efficiency was presented as the slope of the fitted straight line to the
SOA concentration vs OH exposure.






**Table 2.** Correlation coefficients ($R^2$) between POA and SOA UMR mass spectra and
ambient COA resolved by PMF.

|       | CA P[a] | CN P | SR P | PT P | OE P | CA S | CN S | SR S | PT S | OE S | COA[b] |
|-------|------|------|------|------|------|------|------|------|------|------|------|
| CA P  | 1.00 | 0.99 | 1.00 | 0.98 | 0.97 | 0.85 | 0.87 | 0.91 | 0.93 | 0.94 | 0.96 |
| CN P  | 0.99 | 1.00 | 0.99 | 0.99 | 0.99 | 0.89 | 0.90 | 0.94 | 0.96 | 0.96 | 0.95 |
| SR P  | 1.00 | 0.99 | 1.00 | 0.98 | 0.97 | 0.85 | 0.87 | 0.91 | 0.93 | 0.94 | 0.96 |
| PT P  | 0.98 | 0.99 | 0.98 | 1.00 | 0.98 | 0.83 | 0.85 | 0.90 | 0.93 | 0.93 | 0.96 |
| OE P  | 0.97 | 0.99 | 0.97 | 0.98 | 1.00 | 0.86 | 0.88 | 0.93 | 0.95 | 0.96 | 0.94 |
| CA S  | 0.85 | 0.89 | 0.85 | 0.83 | 0.86 | 1.00 | 0.95 | 0.98 | 0.96 | 0.94 | 0.74 |
| CN S  | 0.87 | 0.90 | 0.87 | 0.85 | 0.88 | 0.95 | 1.00 | 0.95 | 0.96 | 0.96 | 0.77 |
| SR S  | 0.91 | 0.94 | 0.91 | 0.90 | 0.93 | 0.98 | 0.95 | 1.00 | 0.99 | 0.97 | 0.83 |
| PT S  | 0.93 | 0.96 | 0.93 | 0.93 | 0.95 | 0.96 | 0.96 | 0.99 | 1.00 | 0.99 | 0.87 |
| OE S  | 0.94 | 0.96 | 0.94 | 0.93 | 0.96 | 0.94 | 0.96 | 0.97 | 0.99 | 1.00 | 0.88 |

[a] CA, CN, SR, PT and OE refer to canola, corn, sunflower, peanut and olive oil.
[b] Lee et al. (2015).



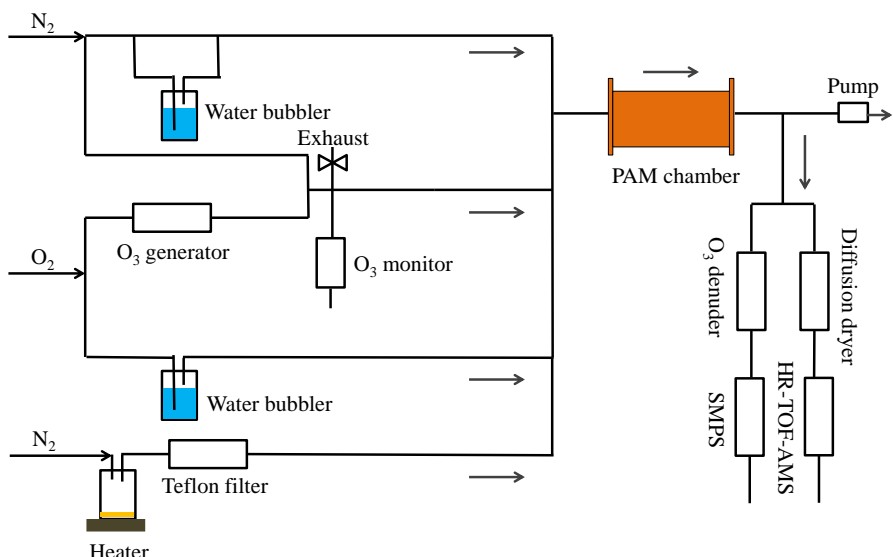


**Fig. 1.** Schematic of the experimental setup.

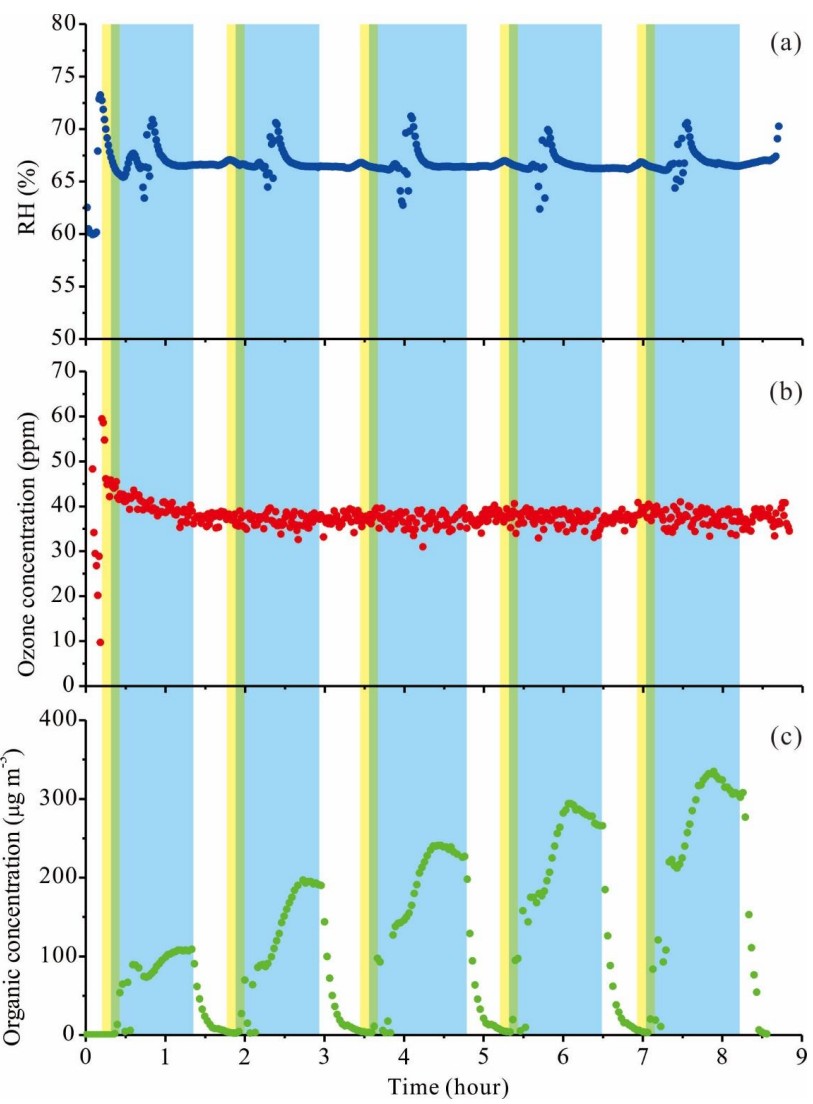


**Fig. 2.** Time series of (a) relative humidity (RH), (b) ozone and (c) organic

concentrations during the aging of gas–phase emissions from heated peanut oil. The

yellow and light blue regions represent the heating oil and OH exposure period,

respectively. The green region is the overlap between heating oil and OH exposure

period.




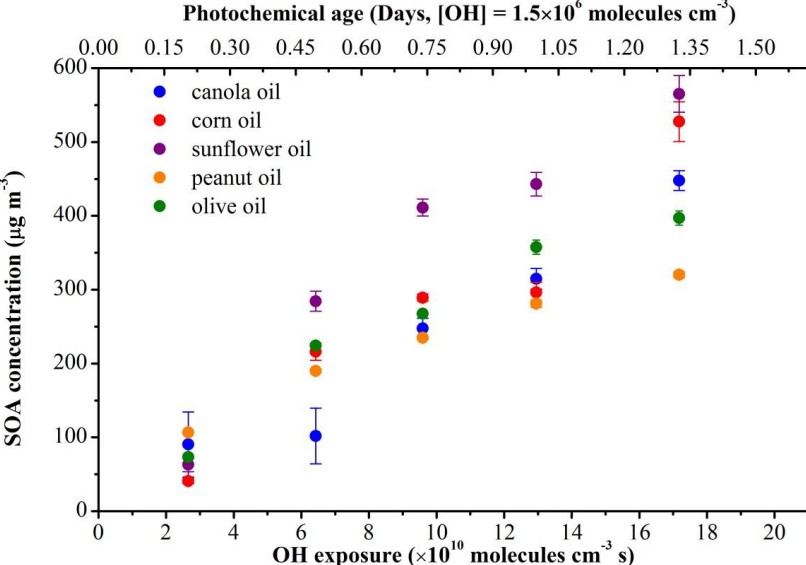


**Fig. 3.** SOA concentration vs. OH exposure and photochemical age in days (at [OH] =
$1.5 \times 10^6$ molecules cm$^{-3}$) during the aging of gas–phase emissions from different heated
cooking oils. Error bars represent the standard deviation (1σ).



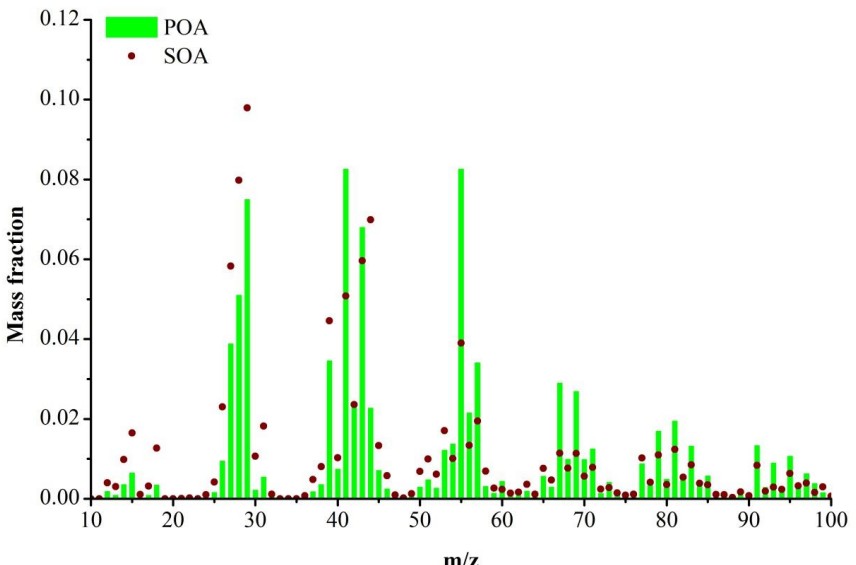


**Fig. 4.** Mass spectra of POA and SOA at an OH exposure of $2.7\times10^{10}$ molecules cm$^{-3}$ s
from heated canola oil.





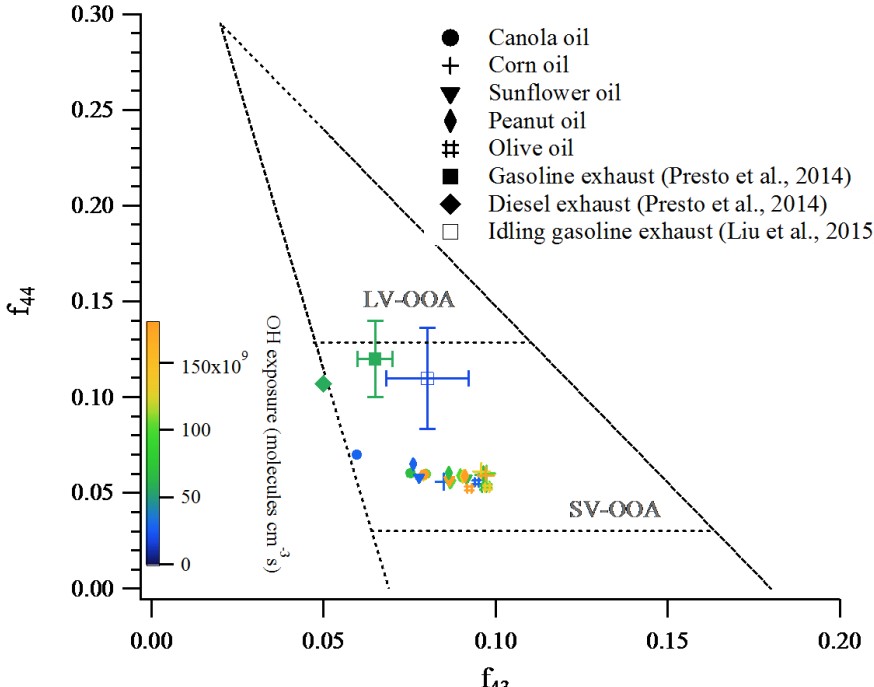

**Fig. 5.** Fractions of total organic signal at m/z 43 ($f_{43}$) vs. m/z ($f_{44}$) from SOA data in this work together with the triangle plot of Ng et al. (2010). SOA data from gasoline (Presto et al., 2014; Liu et al., 2015) and diesel (Presto et al., 2014) vehicle exhaust measured in smog chamber studies are shown. Data from this work and the literature are colored according to OH exposure. Ambient SV–OOA and LV–OOA regions are adapted from Ng et al. (2010).



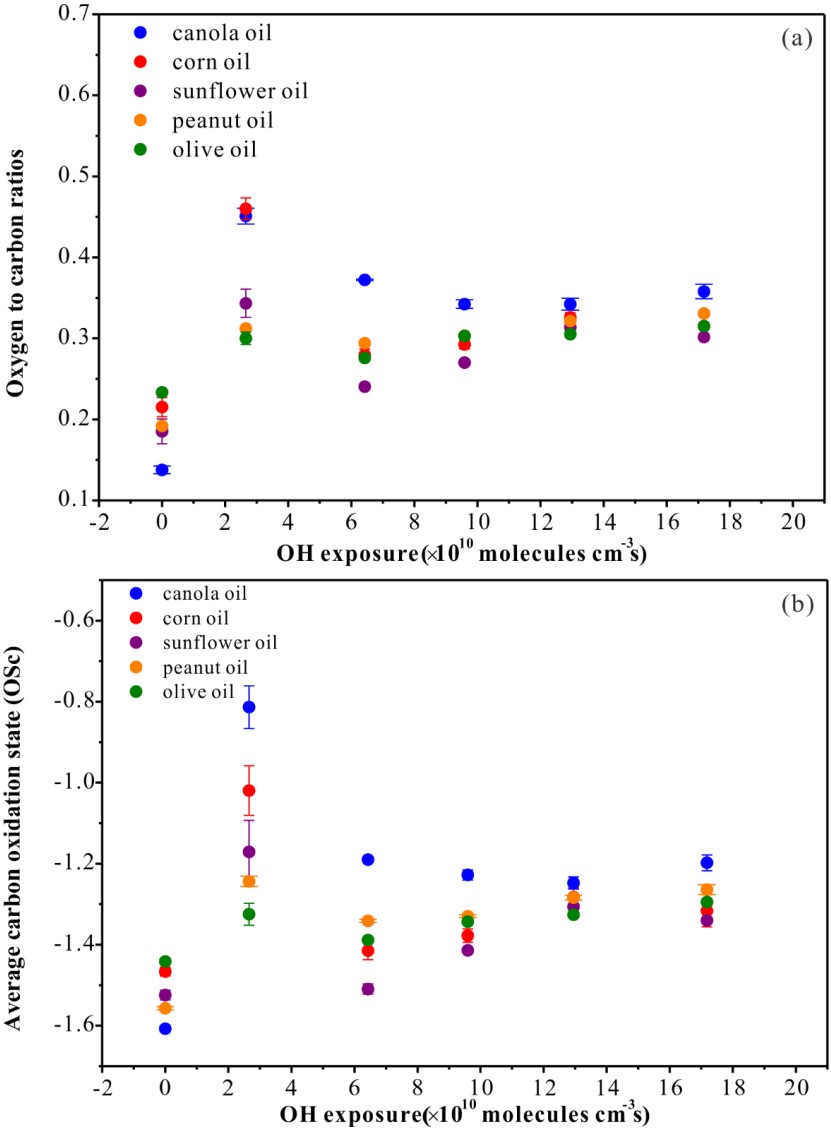


**Fig. 6.** Evolution of (a) oxygen to carbon (O:C) molar ratios and (b) average carbon
oxidation state ($OS_c$) as a function of OH exposure during the aging of gas–phase
emissions from heated different cooking oils, with error bars indicating standard error.
Data at [OH] = 0 represent POA from cooking oils.