# Peer review of "Formation of secondary organic aerosols from gas-phase emissions of heated cooking oils"

_Atmospheric Chemistry and Physics, 2017_

## Referee Comment (RC1) · Anonymous Referee #1 · 8 Feb 2017

This paper shows the production of SOA from oil vapours associated with cooking using a PAM chamber, which is of strong relevance for the modelling and source apportionment of urban particulate matter. While questions persist regarding how atmospherically representative PAM yields are, this paper focuses on the qualitative mass spectral features, comparisons of different oils and speculations on chemical mechanisms, which will no doubt prove highly useful for further studies, so is well within scope for ACP and represents a decent contribution to the science. The paper is generally well written and I recommend publication after consideration of minor comments.

General comments:

I find myself at odds with the conclusion that some of the ambient reported COA in other studies could be secondary in nature because the SOA spectrum shows inconsistencies with the key features normally reported. Specifically, the m/z=41 peak is relatively low compared to m/z=43 and the m/z=44 peak is much higher. By comparison, the POA spectrum is very consistent with the literature. I should note that R2 is not a good metric to compare spectra in the context of discussing PMF outputs because these spectra are derived according to precision-weighted variance, so factors tend to be dictated by the 'strong' variables, i.e. a subset of peaks with the highest signal-to-noise ratios, so it is these features that should really be compared. Furthermore, previously reported diurnal profiles of COA almost universally show a maximum in late evening, which is after dusk in many cases, where oxidation through the OH pathway is not plausible (while ozone and nitrate oxidation of unsaturated bonds will occur, this is not what is investigated here). I think it far more likely that any cooking SOA contribution will be included with one of the OOA factors in a typical urban PMF study. I would recommend that the authors adjust their conclusions accordingly.

All the oils were heated to 220C. While I can recognise the value in experimental consistency, this is possibly beyond the smoke point of some of the oils used here (depending on their grade). Can the authors verify that the oils did not give off visible smoke during the experiments? If they did, this should be added as an important caveat because this would fundamentally alter the emissions profiles.

Specific comments:

Line 67: Allan et al. (2010) did not strictly find the conclusion that cooking oils were a major contribution; this was a speculative explanation for trends within the data. More conventional marker-based studies (e.g. using chromatography) are probably more of more value to support this notion.

Line 111: Was the Teflon transfer line heated? If not, some condensation onto the tube may have occurred. The authors should comment on whether they consider this to be an issue.

Line 125: How were the flow rate and dilution ratio measured/estimated?

Line 163: PR has a number of issues, not least of which is the fact that the oxidant concentrations and simulated timescales in PAM are far in excess of those likely experienced in urban atmospheric and chamber studies. While I am not questioning its usefulness, I would put some caveats in concerning its quantitative merit.

Line 222: It should be noted that the S- and IVOCs will most likely evaporate from the filter if they are not at gas phase saturation, although I concede that given that the aerosol is cooling, it is most likely that they are at saturation in this case. However, an experiment where filtration occurs after dilution may produce different results, so this should be noted.

Line 260: The statement about vegetable oil not being used in commercial kitchens needs further explanation because the vegetable oils studied here are almost universal in many countries. Are the authors referring to Hong Kong specifically? What other oils are in use and why weren't these studied here?

Line 284: Statements are made concerning vapour pressures, but it is all very nonspecific and hand-waving. Some typical saturation vapour pressures (either measured or predicted) for some example compounds should be given.

Line 336: The conclusion regarding animal fat yields should be treated as a speculative inference because while a correlation with omega-6 is found here, its role as a determining factor is not conclusively proven.

Line 343: The 'gas phase' measurements referred to should be specified, e.g. GC-MS.

Supplement: I would argue that both graphs in the supplementary material are of sufficient interest to warrant inclusion in the main article.

Technical comments:

The term 'canola' will not be familiar to all readers worldwide. The alternative name 'rapeseed' should be offered somewhere, by way of explanation.

Line 124: The model of the Dekati dilutor should be given.

Line 139: More information on the diffusion drier must be given. I would note that membranes such as Naphion are known to remove OVOCs. Do the authors have a rough idea what RH the AMS sampled at?

Line 196: It should be noted that the fatty acid vapours studied here are not the specific fats present in the raw oils, but thermal breakdown products of fat lipids.

---

## Referee Comment (RC2) · Anonymous Referee #2 · 28 Mar 2017

**General comments:**

This work reports secondary organic aerosol (SOA) formation by OH radical oxidation of VOCs emitted from heated cooking oil using an oxidation flow reactor approach. Cooking emissions have been recognized as one of the major primary organic aerosol (POA) sources in urban environments. However, SOA formation potential of cooking emissions are largely unknown and hence this study is of great interest to the atmospheric community. Although the observations presented in this manuscript is qualitative due to their experimental limitations, this work clearly demonstrates SOA formation from cooking emissions that provide sufficient insight into future studies. The experiments are well performed in general but some clarifications are required, in particularly the relative importance of other oxidation chemistry in the flow tube reactor. The manuscript is well written and organized. I recommended this manuscript to be pub-
lished in Atmospheric Chemistry and Physics after addressing the specific comments below:

Specific comments:

1. This manuscript focuses on discussing SOA production from OH radical oxidation chemistry. However, a recent modelling study by Peng et al. (2016) has illustrated potential significance of non-OH chemistry in oxidation flow reactors (OFR) for degradation of various SOA precursors. For example, some unsaturated VOCs such as monoterpenes might largely involve in the reaction with ozone depending on operating conditions of OFR. Since this manuscript demonstrates that mono- and poly-unsaturated fatty acids are important SOA precursors from heated cooking oils, it is necessary to comment on the relative importance of non-OH chemistry in the PAM reactor, especially for ozonolysis of unsaturated fatty acids, and how may this related to the observations reported in this manuscript. Control experiments of ozonolysis in the absence of OH radical may provide insight into this issue.

2. Line 88: What was the initial ozone concentration in the PAM reactor?

3. Line 111: Was the Teflon line heated and temperature controlled to minimize wall loss of VOCs? Please clarify.

4. Particle mass determination from SMPS data: Line 131: In general, particle effective density of SOA can be estimated by comparing SMPS data and particle time-of-flight (PToF) measurements from AMS in laboratory experiments. Did the authors conduct this type of estimation to validate their assumption? Furthermore, Figure 1 illustrates that aerosol particles were not dried before SMPS measurements. Please discuss uncertainties of SOA mass calculation due to the presence of aerosol water content.

5. Line 147: What was the collection efficiency (CE) for the SOA produced? Primary cooking organic aerosol is likely oil-like droplets that gives CE approximately equal to 1. Such information can provide insight into the viscosity of SOA produced from
cooking emissions (i.e. particle bouncing on tungsten vaporizer increased with particle viscosity).

6. SOA production rate: I agree that it is reasonable to define SOA production rate as SOA mass produced per minute due to the experimental limitations (i.e. no VOC measurements). However, considering emission rates of VOCs from oils at a given temperature can be different, an additional SOA production rate defined as SOA mass produced per volume of oil evaporated may be able to reduce one experimental variable for interpreting the data (e.g. line 195-196). It is recommended to perform such calculation if the evaporation rate of oils were measured (i.e. volume change of heated cooking oils before and after heating).

7. Lines 213-216: What were the emission rate of POA from heated cooking oils? Please include this information (if available) in the comparison.

8. Lines 256-260: This argument is true if SOA is only a minor contributor to total COA mass because typical mass spectrum of COA factors have high m/z 41-to-m/z 43 ratios which are similar to the POA of heated cooking oil. Furthermore, typical diurnal patterns of COA show a strong peak during dinner time with extremely low OH radical concentration in the atmosphere. Of course, VOCs from cooking can react with other atmospheric oxidants but it is unclear if SOA produced by night time chemistry gives the similar mass spectrum.

**References:**

Peng, Z., Day, D. A., Ortega, A. M., Palm, B. B., Hu, W., Stark, H., Li, R., Tsigaridis, K., Brune, W. H., and Jimenez, J. L.: Non-OH chemistry in oxidation flow reactors for the study of atmospheric chemistry systematically examined by modeling, Atmos. Chem. Phys., 16, 4283-4305, doi:10.5194/acp-16-4283-2016, 2016.

---

## Author Comment (AC1) · 25 Apr 2017

**General comments:**

This paper shows the production of SOA from oil vapours associated with cooking using a PAM chamber, which is of strong relevance for the modelling and source apportionment of urban particulate matter. While questions persist regarding how atmospherically representative PAM yields are, this paper focuses on the qualitative mass spectral features, comparisons of different oils and speculations on chemical mechanisms, which will no doubt prove highly useful for further studies, so is well within scope for ACP and represents a decent contribution to the science. The paper is generally well written and I recommend publication after consideration of minor comments.

**Q1:** I find myself at odds with the conclusion that some of the ambient reported COA in other studies could be secondary in nature because the SOA spectrum shows inconsistencies with the key features normally reported. Specifically, the m/z=41 peak is relatively low compared to m/z=43 and the m/z=44 peak is much higher. By comparison, the POA spectrum is very consistent with the literature. I should note that $R^2$ is not a good metric to compare spectra in the context of discussing PMF outputs because these spectra are derived according to precision-weighted variance, so factors tend to be dictated by the 'strong' variables, i.e. a subset of peaks with the highest signal-to-noise ratios, so it is these features that should really be compared. Furthermore, previously reported diurnal profiles of COA almost universally show a maximum in late evening, which is after dusk in many cases, where oxidation through the OH pathway is not plausible (while ozone and nitrate oxidation of unsaturated bonds will occur, this is not what is investigated here). I think it far more likely that any cooking SOA contribution will be included with one of the OOA factors in a typical urban PMF study. I would recommend that the authors adjust their conclusions accordingly.

R1: We agree with the reviewer that $R^2$ is not a good metric to give a conclusion here. The relevant discussion in the manuscript has been revised accordingly. The following sentences were deleted.

*"suggesting that COA might not be entirely primary in origin"* (Line 31-32).

*"This similarity between the cooking SOA and ambient COA suggests that the COA resolved based on ambient data may be a convolution of POA and SOA, even though vegetable oil may not be the oil commonly used in commercial kitchens."* (Line 294-297).

*"These similarities indicated that ambient COA factors identified by AMS could contain cooking SOA."* (Line 371-372).

**Q2:** All the oils were heated to 220C. While I can recognise the value in experimental consistency, this is possibly beyond the smoke point of some of the oils used here (depending on their grade). Can the authors verify that the oils did not give off visible smoke during the experiments? If they did, this should be added as an important caveat because this would fundamentally alter the emissions profiles.

R2: Visible smoke was only observed during heating olive oil, probably due to the low smoke point of olive oil. According to Klein et al. (2016), olive oil emissions show an increase in larger aldehydes such as nonanal and 2,4-decadienal when the temperature increased from 160 to 220 °C. But for sunflower and canola oil, changing the temperature from 160 to 220 °C do not significantly change the relative composition of their emissions. The following sentences were added to the revised manuscript for clarification.

"Note that visible smoke was observed during heating of olive oil, possibly because the temperature was above the smoke point of olive oil. This high temperature may result in increased emissions of large aldehydes from olive oil, but may not significantly change the relative composition of emissions from other oils with higher smoke points (Klein et al., 2016a)." (Line 122-126).

**Specific comments:**

**Q3:** Line 67: Allan et al. (2010) did not strictly find the conclusion that cooking oils were a major contribution; this was a speculative explanation for trends within the data. More conventional marker-based studies (e.g. using chromatography) are probably more of more value to support this notion.

R3: We agree but at the same time conclusions based on marker studies cannot reveal the relative contribution of cooking oils to PM. A study by Schauer et al. (2002) was added to emphasize the large emissions of organic vapor and aerosols from heating cooking oils.

The sentence "*Allan et al. (2010) found that cooking oils may contribute more to PM than the meat itself in urban areas of London and Manchester.*" has been revised and now reads:

"Allan et al. (2010) postulated that cooking oils may contribute more to PM than the meat itself in urban areas of London and Manchester. Schauer et al. (2002) estimated that cooking seed oils might contribute a significant fraction of lighter n-alkanoic acids such as nonanoic acid in the atmosphere." (Line 67-71).

**Q4:** Line 111: Was the Teflon transfer line heated? If not, some condensation onto the tube may have occurred. The authors should comment on whether they consider this to be an issue.

R4: The Teflon transfer line was not heated. However, the residence time in the transfer line was less than 2 s to minimize wall loss of VOCs. Furthermore, vapors from cooking oils were continuously introduced to the PAM chamber for 10 min to saturate the transfer lines before the UV lamp was turned on. According to Liu et al. (2015), the losses of VOCs in Teflon transfer lines with such a short residence time were less than 5%.

The sentence "*A 2 m Teflon tube was used as the transfer line to minimize wall loss of VOCs.*" has been revised and now reads:

"An unheated 2 m Teflon tube was used as the transfer line. The residence time in the transfer line was less than 2 s, resulting in wall losses of VOCs less than 5% according to Liu et al. (2015). " (Line 127-130).

**Q5:** Line 125: How were the flow rate and dilution ratio measured/estimated?

R5: The original text may have misled the reviewer. The emissions, after passing through a mixing chamber of 36 L, were first diluted by a Dekati diluter (DI-1000, Dekati Ltd, Finland) by a factor of approximately 8. Then 0.15 L min$^{-1}$ of the total diluted flow was introduced to the PAM chamber, achieving a final dilution ratio of

approximately 160. The sentences "*The emissions, after passing through a mixing chamber of 36 L, were introduced to the PAM chamber by a Dekati diluter (DI-1000, Dekati Ltd, Finland) at a flow rate of 0.15 L min$^{-1}$, achieving a final dilution ratio of approximately 160.*" has been revised and now reads:

"The emissions, after passing through a mixing chamber of 36 L, were first diluted by a Dekati diluter (DI-1000, Dekati Ltd, Finland) by a factor of approximately 8. Then 0.15 L min$^{-1}$ of the total diluted flow was introduced to the PAM chamber, achieving a final dilution ratio of approximately 160." (Line 142-146).

**Q6:** Line 163:    PR has a number of issues, not least of which is the fact that the oxidant concentrations and simulated timescales in PAM are far in excess of those likely experienced in urban atmospheric and chamber studies.    While I am not questioning its usefulness, I would put some caveats in concerning its quantitative merit.

R6: We concede that PR may be affected by a number of factors such as the OH exposure and the temperature of the cooking oil. For this study, the OH exposure ranged from $2.7 \times 10^{10}$ molecules cm$^{-3}$ s to $1.7 \times 10^{11}$ molecules cm$^{-3}$ s, equivalent to 0.2–1.3 days of atmospheric oxidation, actually comparable with those in urban atmospheric and chamber studies. As a caution, the following sentence has been added to the revised manuscript.

"Note that PR is highly related to the experimental condition, especially OH exposure and temperature of the cooking oil." (Line 195-196).

**Q7:** Line 222: It should be noted that the S- and IVOCs will most likely evaporate from the filter if they are not at gas phase saturation, although I concede that given that the aerosol is cooling, it is most likely that they are at saturation in this case. However, an experiment where filtration occurs after dilution may produce different results, so this should be noted.

R7: The following sentence has been added.

"SVOCs and IVOCs might not evaporate from the filter given that they might be at saturation as the aerosol was cooled after the emissions." (Line 257-259).

**Q8:** Line 260:    The statement about vegetable oil not being used in commercial kitchens needs further explanation because the vegetable oils studied here are almost

universal in many countries. Are the authors referring to Hong Kong specifically? What other oils are in use and why weren't these studied here?

R8: It is true that vegetable oils are not commonly used in commercial kitchens in Hong Kong. Vegetable oils are widely used in residential cooking. However, this statement has been removed when addressing the Q1.

**Q9:** Line 284: Statements are made concerning vapour pressures, but it is all very nonspecific and hand-waving. Some typical saturation vapour pressures (either measured or predicted) for some example compounds should be given.

R9: The following text was added to the revised manuscript.

"Generally, the presence of additional methylene and aldehyde reduce compound vapor pressure by factors of 3 and 22, respectively (Pankow and Asher, 2007). For example, the vapor pressure of n-tridecanal is approximately 14% of that of n-tridecane at 25 °C, as predicted by the group-contribution model (Pankow and Asher, 2008)." (Line 321-325).

**Q10:** Line 336: The conclusion regarding animal fat yields should be treated as a speculative inference because while a correlation with omega-6 is found here, its role as a determining factor is not conclusively proven.

R10: Softer word was used in the revised manuscript.

"*can be as efficient as vegetable oils in producing SOA*" was changed to "might also produce SOA" (Line 376).

**Q11:** Line 343: The 'gas phase' measurements referred to should be specified, e.g. GC-MS.

R11: Here, 'gas phase' mainly referred to SOA precursors, which can be characterized by GC-MS, PTR-MS and/or other instruments.

"*with gas-phase emissions measured*" was changed to "with gas-phase SOA precursors characterized" (Line 383).

**Q12:** Supplement: I would argue that both graphs in the supplementary material are of sufficient interest to warrant inclusion in the main article.

R12: Both graphs were added to the revised manuscript.

**Technical comments:**

**Q13**: The term 'canola' will not be familiar to all readers worldwide. The alternative name 'rapeseed' should be offered somewhere, by way of explanation.

R13: Rapeseed was added to the revised manuscript when canola oil was mentioned for the first time.

**Q14:** Line 124: The model of the Dekati dilutor should be given.

R14: The model DI-1000 was given in the revised manuscript.

**Q15:** Line 139: More information on the diffusion drier must be given. I would note that membranes such as Naphion are known to remove OVOCs. Do the authors have a rough idea what RH the AMS sampled at?

R15: A silica gel diffusion dryer was used here. It is widely used and will not remove VOCs. The residence time in the dryer was approximated 8 s, sufficient to reduce the RH to less than 30%.

The sentence "*A diffusion dryer was connected to the sampling line to remove water.*" has been revised and now reads:

"A silica gel diffusion dryer was connected to the sampling line to remove water. The residence time in the dryer was approximated 8 s, sufficient to reduce the RH to less than 30%." (Line 159-161).

**Q16**- Line 196: It should be noted that the fatty acid vapors studied here are not the specific fats present in the raw oils, but thermal breakdown products of fat lipids.

R16: Agree. The following sentence was added to the revised manuscript.

"It should be noted that the organic vapors studied here were not the specific fats present in the raw oils, but the thermal breakdown products of fat lipids." (Line 230-232).

References:

Klein, F., Platt, S. M., Farren, N. J., Detournay, A., Bruns, E. A., Bozzetti, C., Daellenbach, K. R., Kilic, D., Kumar, N. K., Pieber, S. M., Slowik, J. G., Temime-Roussel, B., Marchand, N., Hamilton, J. F., Baltensperger, U., Prévôt, A. S. H., and El Haddad, I.: Characterization of Gas-Phase Organics Using Proton Transfer Reaction

Time-of-Flight Mass Spectrometry: Cooking Emissions, Environ Sci Technol, 50, 1243-1250, 10.1021/acs.est.5b04618, 2016.

Liu, T., Wang, X., Deng, W., Hu, Q., Ding, X., Zhang, Y., He, Q., Zhang, Z., Lü, S., Bi, X., Chen, J., and Yu, J.: Secondary organic aerosol formation from photochemical aging of light-duty gasoline vehicle exhausts in a smog chamber, Atmos. Chem. Phys., 15, 9049-9062, 10.5194/acp-15-9049-2015, 2015.

Pankow, J. F., and Asher, W. E.: SIMPOL.1: a simple group contribution method for predicting vapor pressures and enthalpies of vaporization of multifunctional organic compounds, Atmos. Chem. Phys., 8, 2773-2796, 10.5194/acp-8-2773-2008, 2008.

Schauer, J. J., Kleeman, M. J., Cass, G. R., and Simoneit, B. R. T.: Measurement of Emissions from Air Pollution Sources. 4. C1−C27 Organic Compounds from Cooking with Seed Oils, Environ Sci Technol, 36, 567-575, 10.1021/es002053m, 2002.

---

## Author Comment (AC2) · 25 Apr 2017

**General comments:**

This work reports secondary organic aerosol (SOA) formation by OH radical oxidation of VOCs emitted from heated cooking oil using an oxidation flow reactor approach. Cooking emissions have been recognized as one of the major primary organic aerosol (POA) sources in urban environments. However, SOA formation potential of cooking emissions are largely unknown and hence this study is of great interest to the atmospheric community. Although the observations presented in this manuscript is qualitative due to their experimental limitations, this work clearly demonstrates SOA formation from cooking emissions that provide sufficient insight into future studies. The experiments are well performed in general but some clarifications are required, in particularly the relative importance of other oxidation chemistry in the flow tube reactor. The manuscript is well written and organized. I recommended this manuscript to be published in Atmospheric Chemistry and Physics after addressing the specific comments below:

**Specific comments:**

**Q1:** This manuscript focuses on discussing SOA production from OH radical oxidation chemistry. However, a recent modelling study by Peng et al. (2016) has illustrated potential significance of non-OH chemistry in oxidation flow reactors (OFR) for degradation of various SOA precursors. For example, some unsaturated VOCs such as monoterpenes might largely involve in the reaction with ozone depending on operating conditions of OFR. Since this manuscript demonstrates that mono- and poly-unsaturated fatty acids are important SOA precursors from heated cooking oils, it is necessary to comment on the relative importance of non-OH chemistry in the PAM reactor, especially for ozonolysis of unsaturated fatty acids, and how may this related to the observations reported in this manuscript. Control experiments of ozonolysis in the absence of OH radical may provide insight into this issue.

R1: As was also mentioned by Reviewer 1, it should be noted that the organic vapors studied here were not the specific fatty acids present in the raw oils, but the thermal

breakdown products of these fatty acids. According to Klein et al. (2016), emissions of VOCs from heating cooking oils were dominated by saturated and unsaturated aldehydes. In this study, the ratio of $O_3$ exposure to OH exposure ranged from $1.5 \times 10^5$ to $1.9 \times 10^5$, relatively lower than tropospheric values (Schmidt et al., 2014). At this $O_{3exp}/OH_{exp}$, ozonolysis of saturated and unsaturated aldehydes was negligible since the ratios of their ozonolysis rate constants to OH rate constants were in the range of $10^{-9}$ to $10^{-7}$. (Grosjean et al., 1993; Atkinson and Arey, 2003). Thus reactions with $O_3$ played a negligible role in this study. In addition, SOA formation in the absence of OH radicals was also tested. As shown in Fig. 3 in the revised manuscript, during the initial 10 min of heating, the mass concentration of organics was close to the detection limit of the instrument. During these periods of experiments, OH radicals were not present, indicating that ozone chemistry had a negligible influence on SOA formation in this study.

The following sentence has been added to the manuscript.

"Peng et al. (2016) found that non-OH chemistry, especially reactions with $O_3$, may play a role in the oxidation flow reactors for consumption of VOCs. According to Klein et al. (2016a), emissions of VOCs from heating cooking oils were dominated by saturated and unsaturated aldehydes. In this study, the ratio of $O_3$ exposure to OH exposure ranged from $1.5 \times 10^5$ to $1.9 \times 10^5$, relatively lower than tropospheric values (Schmidt et al., 2014). At this $O_{3exp}/OH_{exp}$, ozonolysis of saturated and unsaturated aldehydes was negligible since the ratios of their ozonolysis rate constants to OH rate constants were in the range of $10^{-9}$ to $10^{-7}$. (Grosjean et al., 1993; Atkinson and Arey, 2003). Thus reactions of VOCs with $O_3$ played a negligible role in this study." (Line 99-107).

"During these periods of experiments where OH radicals were not present, we found that ozone chemistry had a negligible influence on SOA formation in this study." (Line 205-207).

**Q2:** Line 88: What was the initial ozone concentration in the PAM reactor?

R2: The ozone concentration in the PAM reactor was adjusted to five different levels, ranging from 0.4 ppm to 2.7 ppm. This sentence has been added to the revised

manuscript (Line 91-92).

**Q3:** Line 111: Was the Teflon line heated and temperature controlled to minimize wall loss of VOCs? Please clarify.

R3: This issue has been addressed in response to Q4 of Reviewer 1.

**Q4:** Particle mass determination from SMPS data: Line 131: In general, particle effective density of SOA can be estimated by comparing SMPS data and particle time-of-flight (PToF) measurements from AMS in laboratory experiments. Did the authors conduct this type of estimation to validate their assumption? Furthermore, Figure 1 illustrates that aerosol particles were not dried before SMPS measurements. Please discuss uncertainties of SOA mass calculation due to the presence of aerosol water content.

R4: The following figure shows the volume distribution of SOA measured by AMS for sunflower oil at an OH exposure of $1.7 \times 10^{11}$ molecules cm$^{-3}$ s. The mode diameter fell in the range of 210 nm to 230 nm. This variation of mode diameter might lead to large uncertainties in calculating the effective density. For instance, the effective density for sunflower oil SOA was estimated to be 1.55±0.28 by comparing SMPS data and PToF measurements from AMS. The density of SOA for other oils was expected to be similar with that for sunflower oil, considering their similar H:C and O:C ratios. Taking considerations of the large uncertainty, the assumption of 1.4 g cm$^{-3}$ is reasonable.

[Figure]

Lambe et al. (2011) investigated the cloud condensation nuclei activity of PAM-generated SOA and found that the hygroscopicity parameter $\kappa_{org}$ was linearly correlated with O:C ratios. Based on their $\kappa_{org}$-to-O:C relationship, we estimated an upper limit of $\kappa_{org}$ to be 0.089 in this study. The overestimate of SOA mass due to water uptake were thus determined to be less than 18%. The following sentences have been added to the revised manuscript.

"Note that particles were not dried prior to SMPS measurements, which might lead to an overestimate of SOA mass due to the uptake of water by organics. Lambe et al. (2011b) investigated the cloud condensation nuclei activity of PAM-generated SOA and found that the hygroscopicity parameter $\kappa_{org}$ was linearly correlated with O:C ratios. Based on their $\kappa_{org}$-to-O:C relationship, we estimated an upper limit of $\kappa_{org}$ to be 0.089 in this study. The overestimate of SOA mass due to water uptake were thus determined to be less than 18% (Petters and Kreidenweis, 2008; Pajunoja et al., 2015)." (Line 171-179).

**Q5:** Line 147: What was the collection efficiency (CE) for the SOA produced? Primary cooking organic aerosol is likely oil-like droplets that gives CE approximately equal to 1. Such information can provide insight into the viscosity of SOA produced from cooking emissions (i.e. particle bouncing on tungsten vaporizer increased with particle

viscosity).

R5: The CE varied from 0.38 to 0.78 in this study. Given that a diffusion dryer was used to remove water, which may change the morphology of SOA, we may not speculate on the viscosity of SOA.

The following sentence has been added to the manuscript.

"The value of CE varied from 0.38 to 0.78 in this study." (Line 171).

Q6: SOA production rate: I agree that it is reasonable to define SOA production rate as SOA mass produced per minute due to the experimental limitations (i.e. no VOC measurements). However, considering emission rates of VOCs from oils at a given temperature can be different, an additional SOA production rate defined as SOA mass produced per volume of oil evaporated may be able to reduce one experimental variable for interpreting the data (e.g. line 195-196). It is recommended to perform such calculation if the evaporation rate of oils were measured (i.e. volume change of heated cooking oils before and after heating).

R6: The reviewer proposed a very good method to constrain the SOA production rate. But the evaporation rate of oils was not measured to perform such calculation in this study. Actually, it is difficult to measure the volume of cooking oils consumed as the oils became stickier after heating. We will adopt this method in the future study.

Q7: Lines 213-216: What were the emission rate of POA from heated cooking oils? Please include this information (if available) in the comparison.

R7: During the measurement of POA, the cooking oils were heated in a pan. We were not sure about the fraction of emissions introduced into the PAM reactor while all emissions were introduced to the PAM for SOA experiments. Thus, only mass spectra and elemental ratios of POA were discussed in the manuscript.

Q8: Lines 256-260: This argument is true if SOA is only a minor contributor to total COA mass because typical mass spectrum of COA factors have high m/z 41-to-m/z 43 ratios which are similar to the POA of heated cooking oil. Furthermore, typical diurnal patterns of COA show a strong peak during dinner time with extremely low OH radical concentration in the atmosphere. Of course, VOCs from cooking can react with other atmospheric oxidants but it is unclear if SOA produced by night time chemistry gives

the similar mass spectrum.

R8: This issue has been addressed in response to Q1 of Reviewer 1.

References:

Atkinson, R., and Arey, J.: Atmospheric Degradation of Volatile Organic Compounds, Chem. Rev., 103, 4605-4638, doi:10.1021/cr0206420, 2003.

Grosjean, D., Grosjean, E., and Williams, E. L.: Rate constants for the gas-phase reactions of ozone with unsaturated alcohols, esters, and carbonyls, Int. J Chem. Kinet., 25, 783-794, doi:10.1002/kin.550250909, 1993.

Klein, F., Platt, S. M., Farren, N. J., Detournay, A., Bruns, E. A., Bozzetti, C., Daellenbach, K. R., Kilic, D., Kumar, N. K., Pieber, S. M., Slowik, J. G., Temime-Roussel, B., Marchand, N., Hamilton, J. F., Baltensperger, U., Prévôt, A. S. H., and El Haddad, I.: Characterization of Gas-Phase Organics Using Proton Transfer Reaction Time-of-Flight Mass Spectrometry: Cooking Emissions, Environ. Sci. Technol., 50, 1243-1250, doi:10.1021/acs.est.5b04618, 2016.

Lambe, A. T., Onasch, T. B., Massoli, P., Croasdale, D. R., Wright, J. P., Ahern, A. T., Williams, L. R., Worsnop, D. R., Brune, W. H., and Davidovits, P.: Laboratory studies of the chemical composition and cloud condensation nuclei (CCN) activity of secondary organic aerosol (SOA) and oxidized primary organic aerosol (OPOA), Atmos. Chem. Phys., 11, 8913-8928, doi:10.5194/acp-11-8913-2011, 2011.

Pajunoja, A., Lambe, A. T., Hakala, J., Rastak, N., Cummings, M. J., Brogan, J. F., Hao, L., Paramonov, M., Hong, J., Prisle, N. L., Malila, J., Romakkaniemi, S., Lehtinen, K. E. J., Laaksonen, A., Kulmala, M., Massoli, P., Onasch, T. B., Donahue, N. M., Riipinen, I., Davidovits, P., Worsnop, D. R., Petäjä, T., and Virtanen, A.: Adsorptive uptake of water by semisolid secondary organic aerosols, Geophys Res Lett, 42, 2015GL063142, doi:10.1002/2015GL063142, 2015.

Peng, Z., Day, D. A., Ortega, A. M., Palm, B. B., Hu, W., Stark, H., Li, R., Tsigaridis, K., Brune, W. H., and Jimenez, J. L.: Non-OH chemistry in oxidation flow reactors for the study of atmospheric chemistry systematically examined by modeling, Atmos. Chem. Phys., 16, 4283-4305, doi:10.5194/acp-16-4283-2016, 2016.

Schmidt, G. A., Kelley, M., Nazarenko, L., Ruedy, R., Russell, G. L., Aleinov, I., Bauer, M., Bauer, S. E., Bhat, M. K., Bleck, R., Canuto, V., Chen, Y.-H., Cheng, Y., Clune, T. L., Del Genio, A., de Fainchtein, R., Faluvegi, G., Hansen, J. E., Healy, R. J., Kiang, N. Y., Koch, D., Lacis, A. A., LeGrande, A. N., Lerner, J., Lo, K. K., Matthews, E. E., Menon, S., Miller, R. L., Oinas, V., Oloso, A. O., Perlwitz, J. P., Puma, M. J., Putman, W. M., Rind, D., Romanou, A., Sato, M., Shindell, D. T., Sun, S., Syed, R. A., Tausnev, N., Tsigaridis, K., Unger, N., Voulgarakis, A., Yao, M.-S., and Zhang, J.: Configuration and assessment of the GISS ModelE2 contributions to the CMIP5 archive, Journal of Advances in Modeling Earth Systems, 6, 141-184, doi:10.1002/2013MS000265, 2014.

Petters, M. D., and Kreidenweis, S. M.: A single parameter representation of hygroscopic growth and cloud condensation nucleus activity, Atmos. Chem. Phys., 7, 1961-1971, doi:10.5194/acp-7-1961-2007, 2007.